# Fast Spot Locating for Low-Density DNA Microarray

**DOI:** 10.3390/s25072135

**Published:** 2025-03-28

**Authors:** MinGin Kim, Jongwon Kim, Sun-Hee Kim, Jong-Dae Kim

**Affiliations:** 1Thermo Fisher Scientific, South San Francisco, CA 94080, USA; hellingford@gmail.com; 2Biomedux, Suwon-si 16226, Republic of Korea; jwkim@biomedux.com; 3Department of Fashion Industry, Incheon National University, Incheon 22012, Republic of Korea; 4School of Software, Hallym University, Chuncheon-si 24252, Republic of Korea

**Keywords:** low-density DNA microarray, spot localization, vectorized programming, square template matching, HPV genotyping

## Abstract

Low-density DNA microarrays are crucial in molecular diagnostics due to their cost-effectiveness and high sensitivity. However, reliable spot localization remains challenging due to positional variations and image artifacts. Traditional intensity-based methods often struggle with weak fluorescence signals. To address this, we propose a rapid spot localization method that combines template matching with point pattern matching, enhanced through vectorized programming and square (box) templates. Vectorized programming accelerated the most time-consuming calculation by 82 times on a PC and was 6000 times faster on a Raspberry Pi compared to a for-loop implementation. While this improvement applies to the vectorized square calculation alone, substantial performance gains were still achieved in the overall process. Additionally, replacing circular templates with square templates resulted in a fourfold reduction in processing time without compromising detection performance. The proposed method effectively reduces computational overhead, making it suitable for high-throughput and resource-constrained applications. The method was validated using HPV genotyping images from commercial DNA microarrays, demonstrating its practical applicability and robust performance in clinical settings.

## 1. Introduction

Low-density DNA microarrays typically contain 20 to 200 probe spots, depending on the diagnostic purpose and target analytes. The shift from traditional mono-assay approaches to multi-assay strategies has significantly improved the quality of practical conclusions by enabling the simultaneous detection of multiple targets. This multi-assay strategy enhances diagnostic accuracy, reduces the risk of false negatives, and provides comprehensive profiling in a single test. These improvements are particularly valuable in clinical settings where identifying co-infections or multiple viral strains is essential for accurate diagnosis and treatment planning [1,2,3,4,5]. Unlike high-density arrays that simultaneously analyze thousands of sequences, low-density microarrays focus on a limited set of markers, making them particularly well suited for small laboratories and clinical environments. Their affordability and minimal infrastructure requirements have promoted their widespread adoption in fields such as infectious disease detection, pharmacogenetics, and oncology [6,7,8]. For example, these microarrays have been effectively applied to diagnose tick-borne pathogens in livestock [7], to genotype antibiotic resistance in Neisseria gonorrhoeae [3], and to perform pharmacogenetic tests aimed at optimizing drug prescriptions [1]. In oncology, they have been instrumental in mutational profiling for colorectal cancer [9] and in enabling noninvasive prenatal screening for conditions like β-thalassemia [8]. Additionally, low-density microarrays are critical for quality control during production, ensuring that probes are reliably immobilized to facilitate accurate diagnostics [10].

Accurate spot localization is paramount in DNA microarray analysis, particularly for low-density arrays where variations in spot center positions may result from factors such as printing pin misalignment, robotic positioning errors, and slide misalignment—issues that encompass both systemic and random components [10].

In practical applications, mismatch between the microarray printer and scanner can cause the entire spot group to shift from its intended position. However, once the spot group’s overall position is determined, the relative positions of individual target probes within the group remain consistent, minimizing the need for further correction.

An additional challenge arises when all target probes are negative, rendering the individual target probe spots undetectable. In such cases, identifying the spot group location becomes challenging. To address this, control probes are strategically positioned at predefined locations that are consistently visible under all conditions. These control probes serve as fixed reference points, enabling the accurate identification of the spot group location even when no positive target spots are present [1,2,6,9].

Furthermore, additional challenges may arise from mechanical drift during scanner operation, the thermal expansion of glass slides, or inaccuracies in the manual placement of slides, which can further distort spot positions. These distortions are particularly problematic in low-density DNA microarrays, where fewer control probes are available for correction, and minor positional deviations may significantly impact results.

Through the combination of spot template matching with point pattern matching, robust spot detection can be achieved even under low-signal conditions. This combined approach offers improved resilience to spot misalignment, ensuring accurate localization even when positional shifts or incomplete spot data are present [11,12,13].

A significant challenge in this methodology arises from common image artifacts, including bright specks, elevated background fluorescence, dust, and scratches [10]. Although the inherent circular shape of spots suggests the use of circular templates for matching [14,15,16], the considerable variability in spot fluorescence necessitates a normalization process—typically implemented through cross-correlation—which in turn requires computing the template’s signal root mean square (RMS) power. This computation can be particularly intensive.

Modern CPUs, featuring multicore architectures, advanced cache hierarchies, and vector processing units, enable significant performance improvements through cache-efficient parallelization and vectorized programming [17,18,19]. These technologies accelerate convolution operations, such as those needed for cross-correlation and RMS power calculations. Given that the primary goal is to localize spots rather than to quantitatively assess probe responses, a box (square) template is proposed as an alternative to circular templates. Box templates allow for separable filtering and facilitate the use of moving average techniques in each direction, thereby substantially reducing computational overhead. When the use of a square template does not compromise detection performance, it can dramatically reduce the time required for spot localization—a critical benefit for high-throughput applications.

In this paper, we present a rapid spot-locating method that leverages vectorized programming and square template matching to overcome the computational challenges associated with normalized correlation. This method is designed to enhance both the efficiency and robustness of spot detection in low-density DNA microarrays, providing critical benefits in resource-constrained and high-throughput diagnostic applications. The proposed approach was further validated using images from the HPV genotyping of patient samples on a commercial DNA microarray, demonstrating its applicability in clinical settings.

## 2. Materials and Methods

### 2.1. Experimental Images

The HPVDNAChip^™^ (Biomedlab Co., Seoul, Republic of Korea) employed in this study adopts a multi-assay approach, allowing the simultaneous detection of multiple HPV types within a single experiment. This design improves diagnostic accuracy and enhances the detection of co-infections and multiple viral strains, which are common in HPV-related diseases, including cervical cancer.

The proposed method was tested using patient images obtained from these chips. The layout of the microarray is depicted in Figure 1. Each slide comprises four chambers, with each chamber allocated for a single-patient sample. To enhance diagnostic reliability, each chamber contains two identical sets of probe spots, composed of four control probes and twenty-two pairs of HPV type-specific oligonucleotide target probes. Each HPV type probe appears twice as a pair, and since two sets exist within each chamber, there are four probes for each HPV type in total. The four human β-globin control probes in each set were used to determine the reference position of a probe set and verify the hybridization process.

Target DNA was extracted from clinical samples, amplified via polymerase chain reaction (PCR), and hybridized onto the chip. During PCR amplification, Cy5 dye was randomly incorporated, allowing hybridization sites to be visualized when scanned with a microarray scanner. For HPV genotyping, the HPVDNAChip^™^ contains 22 type-specific probes—15 targeting high-risk HPV types (16/18/31/33/35/39/45/51/52/56/58/59/66/68/69) and 7 targeting low-risk types (6/11/34/40/42/43/44). Following the manufacturer’s protocol, target HPV DNA was amplified by PCR using HPV-specific and β-globin primers, followed by Cy5-dUTP labeling (NEN Life Science Products, Inc., Boston, MA, USA). The PCR product was hybridized onto the chip at 40 °C for 2 h, and then washed with 3× SSPE and 1× SSPE for 2 min each. Hybridized signals were visualized using a DNA Chip Scanner (Scanarray Lite, GSI Lumonics, Ontario, ON, Canada).

Probes were printed using a microarray spotter equipped with a 100 μm nozzle and subsequently scanned at a 10 μm resolution, yielding a spot radius of approximately 10 pixels. The probe spacing was 300 μm, which translates to a 30-pixel gap between adjacent spots. Consequently, the relative distances of the control spots from the topmost spot were 120, 210, and 300 pixels, respectively.

The scanner captures eight predefined regions on the microarray, each containing a set of spots, and stores the data as 16-bit grayscale images. Each microarray can accommodate up to eight images, with two images assigned per patient. A representative example of the sample image is shown in Figure 1. A total of 1546 images from 773 patient samples were obtained as a dataset in this study.

### 2.2. Template Matching

The control spot pattern template (Figure 2a) can be utilized for the accurate localization of the set of spots within an image. The template provides the highest matching response at the position of the control spots. Within the template, a stable response can be acquired without explicit background elimination by selecting a background region (Bc/Bs) that has the same pixel count as the object area (Oc/Os) and assigning a kernel value of −1.

Since the computational complexity of convolution operations is proportional to the square of the template size, and normalized correlation is required to account for significant variations in spot intensity, directly computing normalized correlation for the entire control spot template (Figure 2a) leads to high computational complexity.

A more computationally efficient alternative involves computing circular spot template matching responses (Figure 2b) and subsequently deriving the point pattern matching, i.e., a constellation matching response (CMR), using the following equation:$$CMR\left(x,y\right)=m\left(x,y\right)+m\left(x,y+d_{1}\right)+m\left(x,y+d_{2}\right)+m\left(x,y+d_{3}\right)$$
where $m\left(\cdot \right)$ represents the spot matching responses at the corresponding positions. And $d_{1}$, $d_{2}$, and $d_{3}$, which are the relative distances of the 2nd, 3rd, and 4th control spots from the 1st control spot, are 120, 210, and 300 pixels, respectively.

On the other hand, if a square template (as shown in Figure 2c) performs well, the computational cost can be significantly reduced further. Unlike circular templates, square templates can be separable in horizontal and vertical directions, allowing for efficient 1D convolution operations [20]. Additionally, moving average filtering can be applied to further optimize the calculations.

### 2.3. Spot Template Matching Response Calculation

The spot template response is calculated using the normalized correlation method:$$m\left(x,y\right)=k\frac{t\left(x,y\right)*i\left(x,y\right)}{\sigma_{i}\left(x,y\right)}$$
where m, i, t, $\sigma_{i}$, ∗, and k are the normalized correlation, input image, template, RMS signal power over the template area, convolution operator, and a constant indicating the template RMS power.

The denominator of the normalized correlation is computed as the cross-correlation between the input image and the template. Since convolution operations are well optimized in most programming languages, we implemented the method using NumPy (2.2.0) and OpenCV (4.11.0) library in Python (3.13.2). These libraries leverage Single Instruction Multiple Data (SIMD), pipelining, and vectorized programming techniques, ensuring efficient computation.

Similarly to the cross-correlation computation, the local RMS power can also be efficiently calculated using convolution:$$\sigma_{i}\left(x,y\right)=\sqrt{i^{2}*P-{\left(i*P\right)}^{2}}$$
where *P* is a kernel with a value of 1 throughout the template domain.

The convolutions of the circular and square templates are performed using OpenCV’s (4.11.0) ‘filter2D’ and ‘boxFilter’ functions, respectively. Both functions are designed with efficient vectorization techniques, and ‘boxFilter’, in particular, fully incorporates separable 1D filtering and moving average calculations, as described earlier [20].

Figure 3 illustrates the steps involved in computing the CMR. Figure 3b shows the spot template matching response derived from Figure 3a, while Figure 3c presents the CMR obtained from the normalized correlation image (Figure 3b). The CMR search area is limited to the upper 200 × 200 pixels, assuming a 1 mm deviation of the spot set (equivalent to 100 pixels).

### 2.4. Verification of Control Spot Localization Performance

Since the spot diameter is 10 pixels, a circular template with a radius of 5 pixels was used to determine the first control spot by maximizing the CMR. This location was overlaid as a circle on the input images for visual inspection. The input images were gamma-corrected (γ = 2) and pseudo-colored using the ‘jet’ colormap to facilitate easier interpretation.

The difference between the first and second peak values in the CMR was used to assess the reliability of peak detection. A relative gain metric was defined to determine the optimal template radius:$$g=\frac{peak_{1}-peak_{2}}{peak_{2}}\times 100$$
where $peak_{1}$ and $peak_{2}$ represent the first and second peak values, respectively, and g is the relative gain of each CMR image.

The algorithm identifies the second peak in the normalized correlation image after detecting the first. Once the first peak is found, the template-sized region surrounding it is set to zero, effectively removing it. The highest value in the resulting image is then identified as the second peak.

The template radius, which defines the half-size of the spot template used in matching, was optimized to enhance detection reliability. A range of candidate radii were evaluated using experimental images. For each radius, the relative gain between the first and second peaks (which measure how much higher the first peak is compared to the second) was calculated for all test images. The minimum relative gain (worst-case scenario) across those images was recorded for each radius. The optimal radius was chosen as the one that maximized this minimum relative gain across all test images. After determining this radius, the spot localization accuracy was reassessed by visually inspecting all images to confirm that the spots were correctly identified.

An excessively small or large radius may cause a significant deviation in the detected spot location from the true spot center. In the dataset used in this study, spots were spaced 30 pixels apart (~300 µm) with a spot diameter of approximately 10 pixels. This allows for a localization error of up to ±10 pixels without compromising the probe detection accuracy.

Figure 4 illustrates the impact of 10-pixel misalignment on control spot localization. Figure 4a shows the case where control spots are perfectly aligned with their true positions, ensuring that the probe judgment grid (designated detection region) precisely overlaps with its corresponding spot. As shown in Figure 4b, even when control spots are misaligned by 10 pixels in both the x and y directions, the spots remain entirely within their respective grid boundaries, ensuring that a 10-pixel offset does not result in misclassification with a neighboring region. This demonstrates that, when classification is based on the spot area appearing brighter than the surrounding region, a 10-pixel localization error remains within a safe margin for correct probe identification.

For non-control spots, probe classification is determined based on whether the spot area appears brighter than the surrounding region, with brighter spots identified as positive and darker spots as negative. Consequently, a 10-pixel misalignment does not significantly impact this classification as long as the spot remains within its designated grid boundary.

Furthermore, when normalized correlation values are used as a probe classification criterion, the peak correlation value within each probe’s designated grid region is employed for classification. As shown in Figure 4c, the grid boundary provides additional tolerance since only the peak value within the grid is considered for classification. Consequently, even if a spot’s true position is offset by 10 pixels, the peak correlation value still falls inside the designated grid area. This reinforces that a 10-pixel localization error does not significantly affect probe classification accuracy under these conditions.

In the square template experiments, the objective was to identify the optimal template size that minimizes localization error while ensuring probe classification reliability. We focused on template sizes that maintained a maximum deviation within 10 pixels to ensure spot locations remained within their designated grid regions. Consequently, template sizes that resulted in localization errors exceeding this threshold were excluded from further analysis to uphold detection robustness.

## 3. Results

### 3.1. Computation Time Analysis

Table 1 presents the average computation times for various vectorized operations required for CMR calculation using experimental images on both a PC (Windows 11) and a Raspberry Pi 4 (Debian 12). The experimental images had a resolution of 520 × 700 pixels, with each pixel stored as a 32-bit floating-point value.

The first data row reports the execution time for computing the square using a for-loop, serving as a baseline reference to emphasize the necessity of vectorized operations, particularly in embedded systems such as the Raspberry Pi.

The second to fourth rows summarize the computation times for point-wise and element-wise image operations, including square computation, square root computation, and image addition. While these operations exhibited similar performance on the PC, the Raspberry Pi required approximately three times longer for square root computation.

The fifth and sixth rows show the computational bottleneck caused by convolution operations. The circle convolution took approximately nine times longer than the square convolution.

The time required to generate the CMR image from the input image and to identify the highest-intensity location is recorded in the “square locating” and “circle locating” rows. The final row presents the time ratio between these two methods, showing that the circle locating step took approximately four times longer than the square locating step.

While the PC was approximately 9 to 10 times faster than the Raspberry Pi, the performance benefits of vectorized computation and the use of a square template for computational efficiency remained consistent across both platforms, as shown in the last row.

The implementation of vectorized operations was crucial for both platforms, as non-vectorized calculations proved to be impractically slow on the Raspberry Pi. The use of a square template resulted in a fourfold improvement in processing speed, and this trend was consistently observed on both systems.

### 3.2. Visual Inspection of Spot Locating

To verify that the detected spot positions were not at the boundaries of the search area, we visually inspected the first detected spot in all experimental images using a circular template.

Figure 5 presents a scatter plot of the detected first spot positions in all experimental images. The pink circle represents the predefined initial position of the first spot, where the control spots were searched over a ±1 mm area. The vertical deviation was measured as 122 pixels (1.22 mm), and the horizontal deviation was 85 pixels (0.85 mm), both within the search range.

Although the initial position was not centered in the distribution, it could be recalibrated during the device calibration process. However, if the proposed fast locating method reliably detects the correct position, recalibration may be unnecessary by allocating the broader search area. Currently, a ± 1mm search area was sufficient for the given dataset.

To further verify the spot detection accuracy, we overlaid a red circle with a radius of 6 pixels at each detected control probe location in all images. For better visualization, images were clipped around the control probe with a margin equal to the search area, gamma-enhanced (γ = 2) and pseudo-colored using the ‘jet’ colormap.

Figure 6 illustrates representative cases, including an image with well-defined control spots and three images where detection could be challenging. The proposed method successfully detected the control spots even in cases where the control spots were faint, one of the control spots had a defect, and the background was highly contaminated due to inadequate washing, which is shown in Figure 6b, Figure 6c, and Figure 6d, respectively.

### 3.3. Optimal Template Size Selection

To determine the optimal template size for spot normalization correlation, we examined the minimum relative gain and the maximum absolute deviation. As shown in Table 2 and Table 3, the safest choice was a circular template with a radius of 6 pixels, as it exhibited the highest minimum relative gain. For circular templates, radii of 6–9 pixels yielded relative gains exceeding 5%, while for square templates, the optimal range was 5–7 pixels.

Figure 7 presents the relative gain histogram when using a radius of 6 pixels. As illustrated in Figure 7, circular templates demonstrated a superior relative gain.

Figure 8 compares the images corresponding to the two lowest and two highest relative gains obtained using circular and square templates. One of the images exhibited the lowest relative gain with the circular template, while the same image showed the second-lowest relative gain with the square template, resulting in a total of only three unique images being presented.

In all three cases, gamma correction with γ = 5 was required to make the control probes visually distinguishable. Despite the control probes being faint, the relative gain remained above 24% when using the square template. These results indicate that the square template is sufficiently stable for processing images of the quality examined in this study.

## 4. Discussion

In this study, we introduced a rapid and robust spot-locating method for low-density DNA microarrays by employing normalized correlation-based template matching combined with constellation matching. By opting for normalized correlation over traditional cross-correlation, we effectively compensated for significant variations in fluorescence intensity, thereby enhancing the accuracy of spot detection. The implementation of an optimal template size determination further ensured the stability of detection across varying data characteristics.

To address the high computational demands associated with normalized correlation, vectorized programming techniques and the use of square templates were integrated into the processing workflow. The application of square templates facilitated separable filtering and the use of moving average algorithms, resulting in an approximately fourfold reduction in processing time. This improvement in efficiency provides critical benefits in resource-constrained and high-throughput diagnostic applications. Furthermore, the compatibility and consistent performance of the proposed method in embedded systems further extends the applicability of the proposed method in point-of-care diagnostic devices. Importantly, while our study considered spots that are approximately 10 pixels in diameter, the proposed method is expected to become even more efficient in higher-resolution imaging scenarios (e.g., 2.5 µm or 5 µm resolution), where processing demands typically increase significantly. This scalability makes our method particularly well suited for advanced microarray analysis requiring fine spatial resolution.

Moreover, our analysis revealed discrepancies between the expression patterns of target spots and control spots within the dataset. This finding underscores the need for reconsidering control probe placement strategies in future DNA microarray designs—such as arranging control probes in a grid formation or positioning them along the array periphery—to better facilitate negative pattern matching through constellation matching. Additionally, the responses derived from constellation matching showed great potential for evaluating both lot quality and individual microarray performance, thereby enhancing overall quality control measures.

## 5. Conclusions

This study demonstrates that the integration of normalized correlation and constellation matching yields a high-speed spot detection method that significantly improves the analytical efficiency of low-density DNA microarrays. The incorporation of vectorized programming and square templates enables substantial reductions in processing time without compromising detection accuracy, making the approach highly suitable for embedded systems and on-site diagnostic applications. While the reported 82-fold and 6000-fold improvements are specific to vectorized calculations, significant gains were also observed in the overall spot-locating process. Furthermore, replacing circular templates with square templates resulted in a fourfold improvement in processing time while maintaining detection performance. Future research should focus on optimizing control probe designs and incorporating additional quality control strategies to further expand the practical applicability of the proposed method.

## Figures and Tables

**Figure 1 sensors-25-02135-f001:**
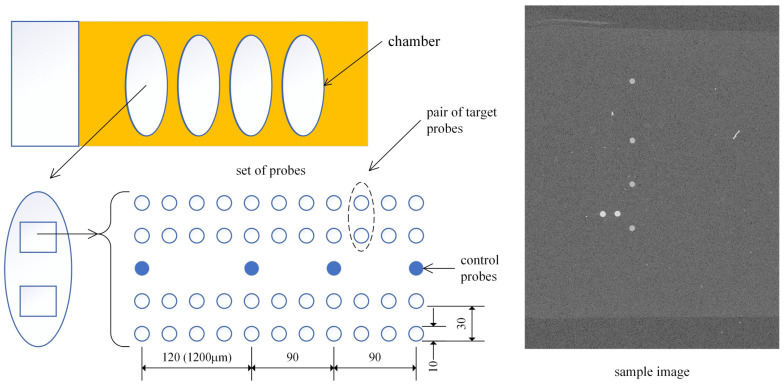
Schematic of HPVDNAChip microarray (sample image is gamma-corrected with γ = 5). The dimension is in pixels (m). Each probe spot is approximately 100 μm in diameter, with a center-to-center distance of 300 μm.

**Figure 2 sensors-25-02135-f002:**
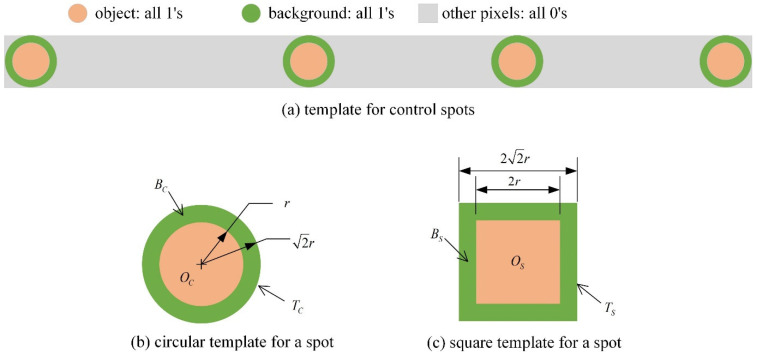
Templates for the control spots and a spot.

**Figure 3 sensors-25-02135-f003:**
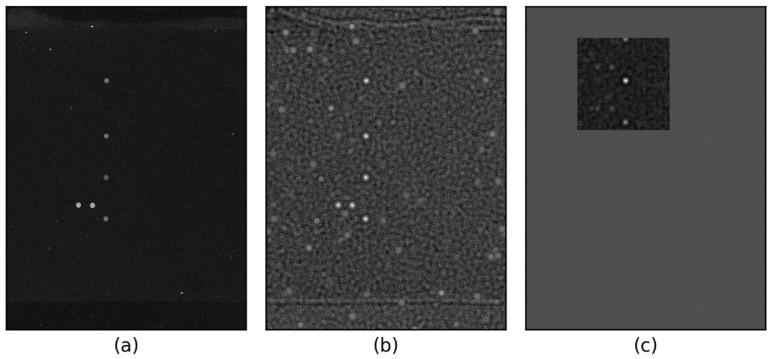
CMR calculation process: (**a**) input image (gamma-corrected, γ=2), (**b**) normalized correlation, (**c**) CMR.

**Figure 4 sensors-25-02135-f004:**
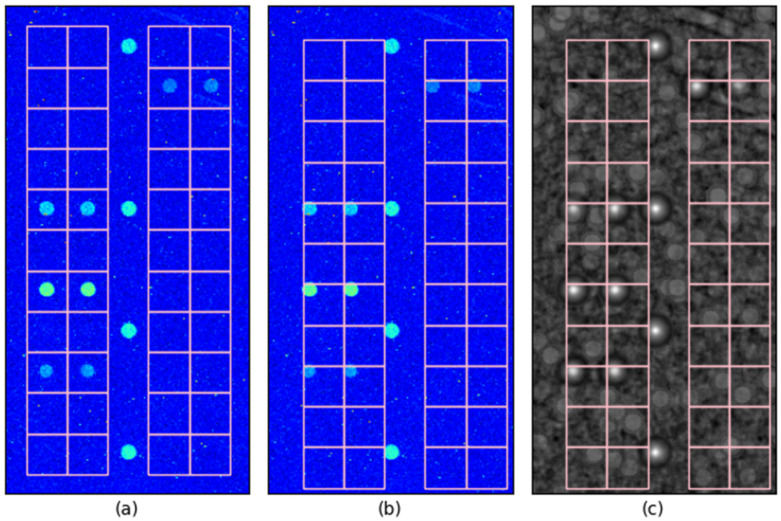
Control spot misalignment tolerance: (**a**) probe judgment grid when spots are precisely located, (**b**) grid misalignment when spots are shifted by 10 pixels in both the x and y directions, (**c**) normalized correlation image overlaid with the misaligned grid.

**Figure 5 sensors-25-02135-f005:**
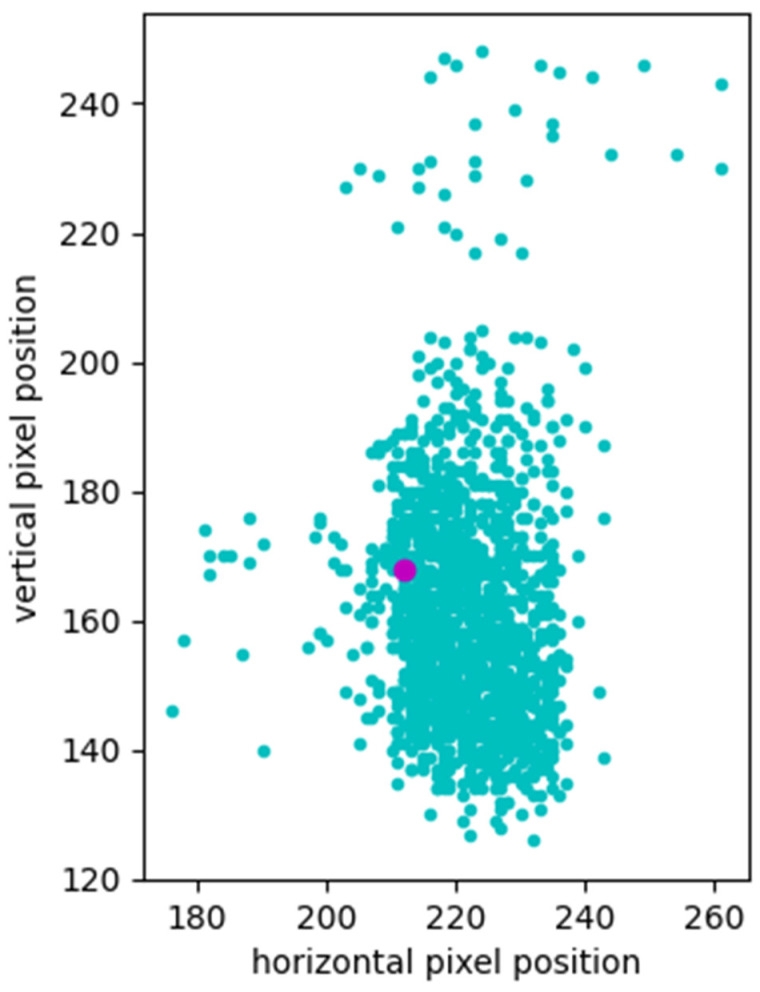
Scatter plot of the first detected spot positions using a circular template (pink circle: predefined initial position).

**Figure 6 sensors-25-02135-f006:**
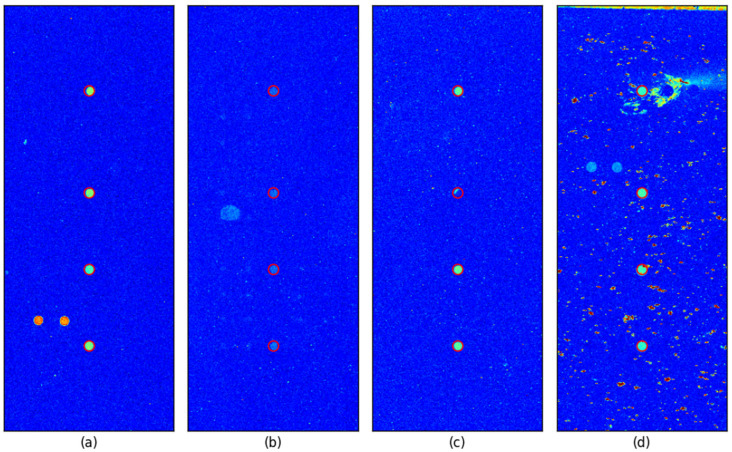
Example images for visual inspection: Each image shows (**a**) clear control spots, (**b**) very faint control spots, (**c**) second control spot defect, (**d**) background contamination due to inadequate washing. (**a**,**d**) represent cases where one HPV type is positive, while (**b**,**c**) correspond to negative cases with no positive target probes present.

**Figure 7 sensors-25-02135-f007:**
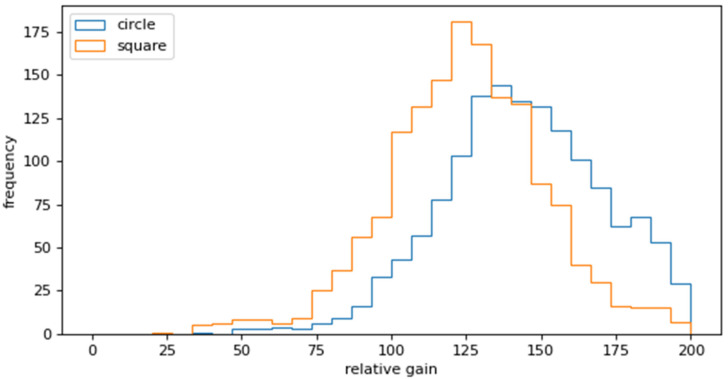
Relative gain histograms when using a radius of 6 pixels.

**Figure 8 sensors-25-02135-f008:**
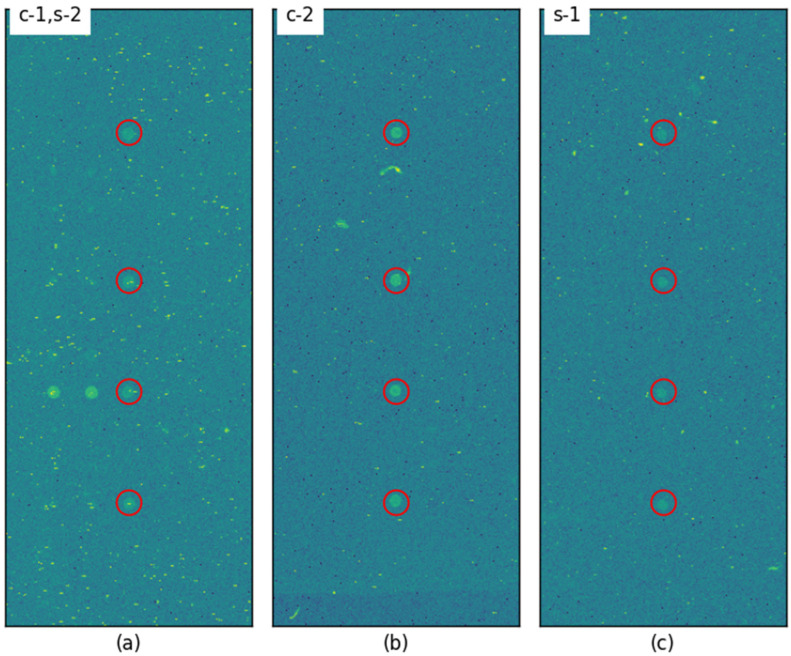
Images with low relative gain: (**a**) the lowest relative gain for the circular template and the second-lowest for the square template, (**b**) the second-lowest relative gain for the circular template, (**c**) the lowest relative gain for the square template (γ = 5, ‘jet’ colormap applied).

**Table 1 sensors-25-02135-t001:** Average computation times required for CMR calculation on a PC and Raspberry Pi 4.

Operation	PC (Windows 11)		Raspberry Pi 4 (Debian 12)	
	Time (μs)	Ratio	Time (ms)	Ratio
for-loop square	4180	82	3540	6000
square	51	1	0.59	1
square root	61	1	1.58	3
image add	69	1	0.58	1
square convolution	428	8	3.03	5
circle convolution	3830	75	36.5	62
square locating	3100	61	26.8	45
circle locating	12,800	251	119	202
circle/square locating		4		4

**Table 2 sensors-25-02135-t002:** Relative gain statistics and maximum header position error for circular templates. The circular template with radius of 6 pixels that shows the highest minimum relative gain is chosen for this method (bolded in table).

Radius (pixels)	5	6	7	8	9
Minimum (%)	0.4	**38.2**	28.9	26.3	8.3
Max. pos. error (pixels)	5.83	0	4.00	5.00	5.00

**Table 3 sensors-25-02135-t003:** Relative gain statistics and maximum header position error for square templates. The circular template with radius of 6 pixels that shows the highest minimum relative gain is chosen for this method (bolded in table).

Radius (pixels)	4	5	6	7	8
Minimum (%)	0.0	8.7	**24.4**	6.3	4.6
Max. pos. error (pixels)	5.83	4.47	3.61	5.00	6.08

## Data Availability

The original contributions presented in this study are included in the article. Further inquiries can be directed to the corresponding authors.

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
