# Peer review of "Fast Spot Locating for Low-Density DNA Microarray"

_sensors, 2025, doi:10.3390/s25072135_

Round 1
Reviewer 1 Report
Comments and Suggestions for Authors
Comments
1) The authors propose to categorize images based on image brightness and darkness in line 215, whether is it based on the contrast of the image or the signal-to-noise ratio of the fluorescent sites?
2) Regarding the localization accuracy, some methods can achieve sub-pixel localization of fluorescence site such as deep learning method, while the localization accuracy in this manuscript is < 10 pixels (Line 222), and the advantages of the method need to be further demonstrated by comparing different methods rather than simply comparing circle and square template.
3) In Fig. 6, the control probe site are effectively recognized, however the target probe sites are not recognized, please explain it.
4) The authors should add what the vertical coordinates represent in Figure 7.
need further polish.
Reviewer 2 Report
Comments and Suggestions for Authors
The manuscript presents a useful improvement of the DNA array technique. The advantages of the proposed approach are clearly demonstrated. The work at whole accords to demands of the Sensors journal, but needs some justifications / revisions.
- The Abstract and Conclusions should present some quantitative data for assessment of possibilities for the proposed approach and the improvements reached by them. Actually even the statement about fast locating in the title is not sufficiently grounded.
- The Introduction should specify typical range for quantity of spots used in low-density DNA microarrays and more specific comments about conclusions that can be made from the assay data (how quality of practical conclusions is improved by the shift from monoassays to multiassays).
- Typically DNA microarray is implemented using supports with the location of initial binding zones that are known to the operators implementing final registration steps of the assay. Thus, the operators typically can apply final image of the spots on initial image that was used for formation of supports. Due to this, the tasks of spots' location should be better grounded in the Introduction: when the necessity of additional recognizing algorithms will arise? The comments actually given at lines 43-48 are very common and do not specify circumstances of such loss of information or distortion of the original localization during the analysis.
- The manipulations with the HPVDNAChips should be better described in the Section 2.1 including sequence and time for each incubation, volumes of the used reactants. It would be better to accomplish the scheme at Fig. 1 by some rulers demonstrating size of the points and distances between them.
- At line 172 the authors consider points with relatively small size (10 pixels), that really causes difficulties in images processing. Please comment how such situation is typical for current microarray practice, or the cases with better resolution of scanning and higher size of the spots in pixels should be considered separately.
- The statement about «more than 75 times longer» convolution given at line 240 is not clearly grounded and needs more detailed consideration of the data that were used for this assessment.
- Similarly to the previous item, the source of the knowledge for the statement about «ten time faster» at line 246 should be indicated. The simple comparison of columns in the Table 1 gives very different proportions between the values for PC and Raspberry PI.
Round 2
Reviewer 1 Report
Comments and Suggestions for Authors
No additional comments.
Reviewer 2 Report
Comments and Suggestions for Authors
The manuscript has been carefully and successfully revised. Its actual version is acceptable for publication.